# Safety and Therapeutic Efficacy of Thymoquinone-Loaded Liposomes against Drug-Sensitive and Drug-Resistant *Acinetobacter baumannii*

**DOI:** 10.3390/pharmaceutics13050677

**Published:** 2021-05-08

**Authors:** Khaled S. Allemailem, Abdullah M. Alnuqaydan, Ahmad Almatroudi, Faris Alrumaihi, Aseel Aljaghwani, Habibullah Khalilullah, Hina Younus, Arif Khan, Masood A. Khan

**Affiliations:** 1Department of Medical Laboratories, College of Applied Medical Sciences, Qassim University, Buraydah 51452, Saudi Arabia; k.allemailem@qu.edu.sa (K.S.A.); aamtrody@qu.edu.sa (A.A.); f_alrumaihi@qu.edu.sa (F.A.); aseel708@hotmail.com (A.A.); 2Department of Medical Biotechnology, College of Applied Medical Sciences, Qassim University, Buraydah 51452, Saudi Arabia; ami.alnuqaydan@qu.edu.sa; 3Department of Pharmaceutical Chemistry & Pharmacognosy, Unaizah College of Pharmacy, Qassim University, Buraydah 51452, Saudi Arabia; shabib79@gmail.com; 4Interdisciplinary Biotechnology Unit, Aligarh Muslim University, Aligarh 202002, India; hinayounus@rediffmail.com; 5Department of Basic Health Sciences, College of Applied Medical Sciences, Qassim University, Buraydah 51452, Saudi Arabia; 4140@qu.edu.sa

**Keywords:** *Acinetobacter baumannii*, thymoquinone, liposome, inflammation, drug-resistant

## Abstract

In the present study, we investigated the activity of free thymoquinone (TQ) or liposomal thymoquinone (Lip-TQ) in comparison to standard antibiotic amoxicillin (AMX) against the drug-sensitive and drug-resistant *Acinetobacter baumannii*. A liposomal formulation of TQ was prepared and characterized and its toxicity was evaluated by analyzing the hematological, liver and kidney function parameters. TQ was effective against both drug-sensitive and drug-resistant *A. baumannii* as shown by the findings of drug susceptibility testing and time kill kinetics. Moreover, the therapeutic efficacy of TQ or Lip-TQ against *A. baumannii* was assessed by the survival rate and the bacterial load in the lung tissues of treated mice. The mice infected with drug-sensitive *A. baumannii* exhibited a 90% survival rate on day 30 post treatment with Lip-TQ at a dose of 10 mg/kg, whereas the mice treated with AMX (10 mg/kg) had a 100% survival rate. On the other hand, the mice infected with drug-resistant *A. baumannii* had a 70% survival rate in the group treated with Lip-TQ, whereas AMX was ineffective against drug-resistant *A. baumannii* and all the mice died within day 30 after the treatment. Moreover, Lip-TQ treatment effectively reduced the bacterial load in the lung tissues of the mice infected with the drug-sensitive and drug-resistant *A. baumannii*. Moreover, the blood of the mice treated with Lip-TQ had reduced levels of inflammation markers, leukocytes and neutrophils. The results of the present study suggest that Lip-TQ may prove to be an effective therapeutic formulation in the treatment of the drug-sensitive or drug-resistant *A. baumannii* infection as well.

## 1. Introduction

*Acinetobacter baumannii* has lately emerged as an important opportunistic pathogen for the skin, bloodstream and urinary tract infections [1]. The use of catheters, endotracheal intubation and immune suppression have been considered the main risk factors associated with *A. baumannii* infections [2]. Currently, it poses major health problems due to a sudden upsurge in the numbers of antibiotic-resistant *A. baumannii* isolates that are mounting a serious challenge to clinicians and researchers. Lately, carbapenem-resistant isolates of *A. baumannii* have surfaced in many parts of the world [3]. Aly et al. demonstrated that among all the *A. baumannii* isolates reported from 2006 to 2008 at King Abdulaziz Medical City Hospital in Riyadh, Saudi Arabia, 79.1% exhibited resistance to imipenem and 92.1%—to meropenem [4]. In another study, Saeed et al. showed that 93% of all the *A. baumannii* isolates exhibited resistance to piperacillin/tazobactam, 92%—to ciprofloxacin, 96%—to amoxicillin/clavulanic acid [5]. Overall, these reports warn us that *A. baumannii* is developing antibiotic resistance very fast and is going to pose a big threat to humanity in the coming years. It adopts various drug resistance mechanisms, including the development of efflux pumps, beta-lactamase enzymes, changes in the outer membrane and penicillin-binding proteins [6].

The seeds of *Nigella sativa*, commonly known as black seed or black cumin, have been widely used in the Arabic, Ayurveda, Chinese and Unani systems of medicine [7,8,9] because *N. sativa* has numerous nutritional and medicinal benefits against cancer, diabetes, asthma, hypertension and eczema [7,10]. The black seed contains many important phytoconstituents, including thymoquinone, thymohydroquinone, dithymoquinone, carvacrol, thymol, nigellicine, nigellidine, p-cymene, etc. [8]. Thymoquinone (TQ), a chief constituent of black seed, targets multiple cell signaling pathways and has potential therapeutic implications in many diseases (Figure 1) [7]. Importantly, TQ exhibited the broad-spectrum antimicrobial activity against various pathogens, including bacteria, fungi and viruses [7]. Moreover, TQ effectively inhibited biofilm formation in bacterial and fungal pathogens [11,12,13]. Interestingly, TQ inhibited the functioning of the drug efflux pumps in bacteria, including *Pseudomonas aeruginosa*, *Staphylococcus aureus* and *Bacillus*
*cereus* [14]. In addition to their activity against bacteria and fungi, black seed and TQ also exhibited activity against many viruses, including mouse hepatitis virus (MHV)-A59, cytomegalovirus, hepatitis C virus (HCV) and avian influenza virus [15].

Liposomes have been used as effective drug and vaccine delivery systems in model animals [16]. Moreover, the pegylation of lipid nanoparticles has been found to be more effective in selective organ targeting of antigens [17]. Because of its poor solubility in aqueous media, the therapeutic potential of TQ has not been adequately utilized. Incorporation of TQ in drug delivery systems has resulted in greater effectiveness and bioavailability [18]. TQ-incorporated liposomes inhibited the proliferation of breast cancer cells [16], whereas TQ–gold niosomes were effective in breaking drug resistance in MCF7 breast cancer cells [19]. Earlier, we demonstrated that liposomal TQ effectively combated fluconazole-resistant *Candida albicans* in a mouse model [20]. The present study aimed to develop a liposomal formulation of TQ (Lip-TQ) that can be effective in fighting both drug-sensitive and drug-resistant *A. baumannii*. Here, we determined both in vitro and in vivo activity of Lip-TQ against the drug-sensitive and drug-resistant *A. baumannii*. The results demonstrated that treatment with Lip-TQ was effective in curing the mice infected with the drug-sensitive and drug-resistant *A. baumannii*.

## 2. Materials and Methods

### 2.1. Materials

Liposome-grade cholesterol and 1,2-dipalmitoyl-Sn-glycero-3-phosphocholine (DPPC) were obtained from Avanti Polar Lipids (Alabaster, AL, USA). Nutrient agar and nutrient broth were procured from HiMedia (Mumbai, India). Thymoquinone was bought from Sigma-Aldrich (St. Louis, MO, USA). A mouse C-reactive protein (CRP) ELISA kit was obtained from Sigma-Aldrich (St. Louis, MO, USA). A mouse procalcitonin (PCT) ELISA kit was obtained from Cusabio Biotech (Carlsbad, CA, USA).

### 2.2. Test Strain of Acinetobacter baumannii

The drug-resistant strain of *A. baumannii* (ATCC 19606) was acquired from the American Type Culture Collection (ATCC, Manassas, VA, USA), whereas the drug-sensitive isolate was obtained from the Department of Microbiology, King Fahad Specialist Hospital, Buraydah, Saudi Arabia. Both strains were maintained on Mueller–Hinton agar (MHA) culture media plates.

### 2.3. Screening of A. baumannii by Automated Susceptibility Testing

The susceptibility testing of *A. baumannii* was performed according to the EUCAST guidelines similar to the CLSI 2016 guidelines [21,22]. Briefly, an *A. baumannii* suspension was adjusted to the turbidity equal to 0.5 McFarland standard. Tryptone soy agar (TSA) plates (Oxoid Limited, Basingstoke, UK) were prepared and each plate was inoculated with 60 μL of the *A. baumannii* suspension. Antibiotic discs (Oxoid Limited) were placed on TSA plates just after plate streaking. Consequently, the plates were kept in the incubator at 37 °C for 24 h to observe the drug susceptibility testing. The following antibiotic discs were used to test the susceptibility of *A. baumannii*: ampicillin (10 μg), chloramphenicol (30 μg), cefotaxime (30 μg), ceftriaxone (30 μg), cotrimoxazole (25 μg), cephalothin (30 μg), clindamycin (2 μg), nitrofurantoin (300 μg), norfloxacin (10 μg), ceftazidime (30 μg), tobramycin (10 μg), ciprofloxacin (5 μg), nalidixic acid (30 μg), imipenem (10 μg) and piperacillin (100 μg).

### 2.4. Determination of In Vitro Activity of TQ against A. baumannii

The activity of TQ against *A. baumannii* was determined by the agar well diffusion method and the broth macrodilution susceptibility method. Nutrient agar plates were prepared and seeded with the *A. baumannii* inoculum. The wells of 8-mm diameter were created and 50 µL containing 25, 50 and 100 µg of TQ or amoxycillin (AMX) were loaded in each well. The plates were incubated at 37 °C for 24 h. The activity of TQ against *A. baumannii* was determined by measuring the growth inhibition zone.

The minimum inhibitory concentration (MIC) of TQ against *A. baumannii* was determined by following the guidelines of the Clinical and Laboratory Standards Institute (CLSI) [21]. Various concentrations of TQ or AMX (0.125 to 128 µg/mL) were used to determine their MIC against *A. baumannii*. *A. baumannii* was cultured in nutrient broth at 37 °C. After 24 h, the cells were centrifuged and *A. baumannii* cells were suspended in nutrient broth to 0.5 McFarland turbidity and added to tubes containing various concentrations of TQ or AMX. The tubes were incubated at 37 °C for 24 h. The lowest concentration of TQ or AMX at which *A. baumannii* did not exhibit any turbidity was considered as the MIC of the drug.

The antibacterial activity of TQ or AMX was also observed through microscopic analysis. In a 24-well sterile culture plate, 1 × 10^6^ CFUs of *A. baumannii* were added to 1 mL nutrient broth at 37 °C. After 4 h, TQ or AMX at the concentrations of 1, 2.5 and 5 µg/mL were added to the well containing the bacteria, whereas the well containing the vehicle was considered the negative control. After 24 h of incubation, the wells were washed with the phosphate-buffered saline (PBS) and observed under the microscope.

### 2.5. Time-Kill Assay

A time-kill assay was performed with 5 × 10^5^ CFU/mL of *A. baumannii* with 1, 2, 4 and 8 µg/mL of TQ or AMX as described earlier [23]. After overnight culture, *A. baumannii* was diluted with the Mueller–Hinton broth (MHB). This culture was incubated at 37 °C until the log phase of growth was achieved. The *A. baumannii* suspension (5 × 10^5^ CFUs) was transferred to flasks containing 20 mL of the MHB and the abovementioned amounts of TQ or AMX. Samples (0.5 mL) were taken in duplicates at the baseline and after 1, 3, 6, 12 and 24 h. The samples were centrifuged at 12,000× *g* for 15 min and reconstituted with sterile PBS to the original volume to minimize any drug carryover effect. The numbers of bacteria were quantified by plating the serial dilutions onto Mueller–Hinton agar (MHA) plates and incubating them at 37 °C. The bacterial density of each sample was determined by counting the CFUs and the limit of detection was considered 100 CFU/mL.

### 2.6. Preparation of TQ-Loaded Liposomes

DPPC, cholesterol and TQ were dissolved into the mixture of methanol and chloroform (1:1 *v*/*v*) as described earlier [24]. The molar ratio of lipids and TQ was kept 10:1. All the ingredients were taken in a round bottom flask and the solvents were evaporated using a rotary evaporator to form a thin lipid film at 37 °C. It was subjected to vacuum to remove traces of the solvents. The film was hydrated with an appropriate amount of sterile PBS at 37 °C. The suspension was sonicated to prepare TQ-loaded liposomes. Free TQ was separated by centrifuging the suspension at 25,000× *g* for 30 min.

### 2.7. Characterization of Liposomes

Liposomes were passed through a membrane (100 nm) using a mini-extruder device procured from Avanti Polar Lipids (Alabaster, AL, USA). The size and shape of liposomes were determined using transmission electron microscopy (Hillsboro, OR, USA). The samples were dropped on a carbon-coated copper grid with consequent staining and drying. The images were taken using a high-resolution TEM operating at 200 kV [25]. The polydispersity index (PDI) of TQ-loaded liposomes was also analyzed using a Malvern Nano Zetasizer (Malvern Instruments, Southborough, MA, USA) employing the dynamic light scattering (DLS) technique as described earlier [25].

### 2.8. Determination of TQ Entrapment Efficiency (EE) in Liposomes

The quantity of TQ entrapped in liposomes was estimated by taking the absorbance at 330 nm [24]. An aliquot of liposomal TQ (100 µL) was disrupted in DMSO and the released TQ was quantified using the standard curve of TQ. The percent EE of TQ was calculated by estimating the entrapped TQ out of the total quantity of TQ added to the lipids.
% EE of TQ = (liposome-entrapped TQ / total TQ) × 100(1)

### 2.9. Determination of Stability of Liposomes and Release Kinetics of TQ

The stability of Lip-TQ was determined at 37 °C as described earlier [26]. A sample (1 mL) of the liposomal suspension was placed in the cellulose dialysis tubing in 25 mL of sterile distilled water in a water bath maintained at 37 °C with constant slow stirring. At various time points (1, 2, 3, 6, 12 and 24 h), 1 mL of volume was taken from the beaker and 1 mL of fresh distilled water was added to maintain the original volume. The absorbance of the withdrawing aliquot was measured at 330 nm to determine the leakage of TQ.

The release kinetics of TQ from the liposomal formulation was determined by incubating the mixture consisting of one volume of Lip-TQ with nine volumes of human serum. The reaction mixtures were incubated for 1, 2, 3, 6, 12 and 24 h at 37 °C as described earlier [26]. The mixture was centrifuged at 12,000× *g* for 15 min. The amount of TQ released from Lip-TQ into the serum was estimated as a percentage of total TQ present in the liposomes initially added to the serum.

### 2.10. Mice

Male Swiss mice (12-week-old) were taken from the animal house facility of the College of Applied Medical Sciences, Qassim University, Saudi Arabia. The experiments were conducted according to the guidelines and regulations of the animal ethics committee of the College of Applied Medical Sciences, Qassim University. Before the infection, the mice were anesthetized by injecting a mixture of ketamine (90 mg/kg) and xylazine (10 mg/kg) intraperitoneally. The infected mice were observed every day for their mortality and morbidity during the study period. This study was approved by the animal ethics committee (approval number 5604-cams1-2019-2-2-I, 19 December 2019).

#### 2.10.1. Determination of TQ Toxicity in Mice

In order to determine toxicity, various doses of free TQ and Lip-TQ were administered to mice intraperitoneally for seven consecutive days. For in vivo administration, TQ was dissolved in DMSO and diluted with normal saline to have 1% DMSO in the final solution. The mice were observed for any morbidity and mortality during this period. The mice were randomly divided into the following groups and each group contained six mice:(1)Saline(2)Sham liposomes(3)Free TQ—1 mg/kg(4)Free TQ—10 mg/kg(5)Free TQ—20 mg/kg(6)Free TQ—40 mg/kg(7)Lip-TQ—1 mg/kg(8)Lip-TQ—10 mg/kg(9)Lip-TQ—20 mg/kg(10)Lip-TQ—40 mg/kg

#### 2.10.2. Analysis of Hematological and Biochemical Parameters

On day 8, blood was collected from the mice of each group through a retroorbital puncture. The erythrocyte, leukocyte and platelet counts were determined. Moreover, the levels of AST (aspartate transaminase) and ALT (alanine transaminase) as liver inflammation markers were determined, whereas the values of creatinine and blood urea nitrogen (BUN) were determined as kidney function parameters [27].

#### 2.10.3. Standardization of *A. baumannii* Infection in Mice

*A. baumannii* (ATCC 19606) were grown in a nutrient broth at 37 °C for 24 h. *A. baumannii* cells were centrifuged at 5000 rpm for 15 min at 4 °C. The cell pellet was washed twice with normal saline. The bacterial cells were counted and mice in various groups were infected with 1 × 10^6^, 5 × 10^6^, 1 × 10^7^, 5 × 10^7^, 1 × 10^8^ CFUs of *A. baumannii* intravenously.

#### 2.10.4. Mouse Model of *A. baumannii* Infection

After the standardization of the infection dose, each mouse was intravenously infected with 1 × 10^7^ CFUs of *A. baumannii*.

#### 2.10.5. Treatment of *A. baumannii*-Infected Mice with TQ Formulations

The efficacy of free or Lip-TQ (1, 5 and 10 mg/kg) was determined against *A. baumannii* in Swiss mice. After 12 h of the *A. baumannii* infection, the mice were treated with single daily doses of the TQ formulation for seven successive days. The mice were divided into eight groups and each group contained ten mice: (1) saline, (2) sham Lliposomes, (3) free TQ (1 mg/kg), (4) free TQ (5 mg/kg), (5) free TQ (10 mg/kg), (6) Lip-TQ (1 mg/kg), (7) Lip-TQ (5 mg/kg), (8) Lip-TQ (10 mg/kg). The mice were observed for a period of 30 days after the *A. baumannii* infection.

#### 2.10.6. Determination of the Bacterial Load in Lung Tissues

The effectiveness of TQ formulations was assessed by analyzing the bacterial load in lung tissues as described earlier [27]. Three mice from each group were sacrificed on day 5 post-treatment and equally weighed portions of the lung tissue were homogenized in sterile normal saline. After appropriate dilution in sterile normal saline, 100 μL of the tissue homogenate were spread on nutrient agar plates. The plates were incubated at 37 °C for 24 h to observe the growth of *A. baumannii*. The number of CFUs of *A. baumannii* in lung tissues was calculated using multiplication by the dilution factor.

#### 2.10.7. Determination of Leukocytes and Inflammation Markers

On day 5 post-treatment, the total leukocyte and neutrophil counts were analyzed in the blood of mice using a hematology autoanalyzer [27]. The C-reactive protein (CRP) and procalcitonin (PCT) are commonly used inflammatory biomarkers to diagnose bacterial infections. The CRP and PCT levels were determined in blood samples using ELISA.

#### 2.10.8. Statistical Analyses

The survival rate of the mice was analyzed using the Kaplan–Meier curve with the log-rank chi-squared test. The data of the bacterial load were analyzed by one-way ANOVA followed by the Bonferroni post-hoc test using the GraphPad Prism software, version 5.0 (La Jolla, CA, USA).

## 3. Results

### 3.1. Antibiotic Susceptibility Pattern of A. baumannii

The findings of the antibiotic disc susceptibility testing showed that drug-sensitive *A. baumannii* exhibited sensitivity to most of the antibiotics, including ampicillin (AP), ciprofloxacin (CIP), chloramphenicol (C), ceftriaxone (CRO), cefotaxime (CTX), penicillin (PG), cefoxitin (FOX), cephalothin (KF), clindamycin (CD), norfloxacin (NOR), nitrofurantoin (NI), tobramycin (TN), ceftazidime (CAZ), imipenem (IMI) and piperacillin (PRL) (Figure 2A). *A. baumannii* (ATCC 19606) has been reported to exhibit resistance against various antibiotics [28]. Figure 2B shows that *A. baumannii* (ATCC 19606) was resistant to most of the antibiotics, including AP, C, CRO, CTX, cotrimoxazole (TS), PG, CD, FOX, NI, KF, CAZ and aztreonam (ATM), whereas it exhibited susceptibility to CIP, nalidixic acid (NA), IMI and PRL (Figure 2B).

### 3.2. In Vitro Activity of TQ against A. baumannii

TQ demonstrated very strong activity against the drug-sensitive and drug-resistant *A. baumannii* (ATCC 19606) as well. The wells containing 25, 50 and 100 μg of TQ exhibited 15-, 24- and 42-mm inhibition zones against drug-sensitive *A. baumannii* (Figure 3A), whereas the wells containing the same amounts of TQ exhibited 12-, 18- and 25-mm growth inhibition zones in the plates streaked with drug-resistant *A. baumannii* (Figure 3B). On the other hand, amoxicillin (AMX) demonstrated its strong activity against the drug-sensitive, but not against the drug-resistant *A. baumannii* (Figure 3C,D). The wells containing 25, 50 and 100 μg of AMX exhibited 30-, 35- and 50-mm growth inhibition zones against drug-sensitive *A. baumannii* (Figure 3C), whereas drug-resistant *A. baumannii* (ATCC 19606) did not exhibit any susceptibility to AMX at the given concentrations (Figure 3D).

The activity of TQ or AMX against the drug-sensitive and drug resistant *A. baumannii* was analyzed using microscopic analysis. The results revealed that TQ had strong activity against the drug-sensitive and drug-resistant *A. baumannii*, whereas AMX was effective against the drug-sensitive, but not the drug-resistant *A. baumannii*. TQ exhibited a dose-dependent activity against the drug-sensitive and drug-resistant *A. baumannii* as well (Figure 4A,B). However, AMX was highly effective against drug-sensitive *A. baumannii*, but it was ineffective against drug-resistant *A. baumannii* (Figure 4C,D).

The MIC of TQ was determined by observing the turbidity of growth of *A. baumannii*. The MIC values for the drug-sensitive and drug-resistant *A. baumannii* were found to be 2 and 5 µg/mL, respectively, whereas the MIC of AMX against drug-sensitive *A. baumannii* was found to be 1 µg/mL.

### 3.3. Activity of TQ against the Drug-Sensitive and Drug-Resistant A. baumannii According to Time-Kill Studies

The bactericidal effect of TQ against the drug-sensitive and drug-resistant *A. baumannii* was studied using time-kill studies. The extent of *A. baumannii* killing was determined by a decrease in the number of CFUs at different timepoints. The findings of the present study demonstrated that there was time-dependent and dose-dependent activity of TQ against both drug-sensitive and drug-resistant *A. baumannii* (Figure 5A,B). TQ at all concentrations exhibited bactericidal effects against both drug-sensitive and drug-resistant *A. baumannii* at the 24-h timepoint. However, TQ at the concentrations of 2 and 4 µg/mL reduced the relative amount of drug-sensitive *A. baumannii* by ≥ 3 log_10_ CFUs/mL (Figure 5A). TQ, at the concentrations of 1 and 2 µg/mL decreased the bacterial count by 60% and 93%, whereas TQ at the concentrations of 4 and 8 µg/mL killed 99.99% of *A. baumannii* (Figure 5A). On the contrary, drug-resistant *A. baumannii* exhibited less susceptibility to TQ, particularly at lower concentrations (Figure 5B). TQ at 1 and 2 µg/mL killed 44.6% and 75.7% of the original bacterial inoculum, whereas 4 and 8 µg/mL of TQ killed 99.4% and > 99.99% of drug-resistant *A. baumannii* (Figure 5B).

AMX demonstrated strong activity against drug-sensitive *A. baumannii* (Figure 5C). AMX at the concentrations of 2 and 4 µg/mL reduced the relative amount of drug-sensitive *A. baumannii* by ≥ 3 log_10_ CFU/mL (Figure 5C). AMX at the concentrations of 1, 2 and 4 µg/mL killed more than 99% of *A. baumannii* (Figure 5C). On the other hand, AMX was not effective against drug-resistant *A. baumannii* at the given concentrations (Figure 5D). 

### 3.4. Characterization of TQ Liposomes

The PDI value indicates the homogeneity of nanoparticles between 0 and 1. The PDI value of the TQ-loaded liposomes was found to be 0.212 ± 0.006. The lower PDI value of 0.212 indicates more uniformity of the liposomal TQ formulation. The zeta potential of thymoquinone liposomes was found to be 2.1 mV. The mean size of TQ liposomes was about 128 nm as determined by the TEM and DLS (Figure 6A,B). The percent EE of TQ was found to be about 90%.

### 3.5. Stability of TQ-Encapsulated Liposomes and Release Kinetics of TQ

The stability of Lip-TQ was determined in deionized water. As shown in Figure 7A, the release of TQ was found to be 3.76% and 4.2% at 12- and 24-h timepoints, respectively (Figure 7A). This small leakage of TQ from the liposomes suggests that the drug strongly interacted with the liposomes. The TQ-loaded liposomes were found to be quite stable in the presence of serum. The percent release of TQ from liposomes was found to be 16 ± 0.53 and 21 ± 0.87 after 12 and 24 h of incubation, respectively (Figure 7B).

### 3.6. Administration of Free TQ or Lip-TQ Did Not Induce any Significant Toxicity

Free TQ or Lip-TQ at the doses of 1, 10, 20 and 40 mg/kg were administered to mice. The mice in the groups injected with the highest doses of free TQ or Lip-TQ (40 mg/kg) did not exhibit any mortality (Table 1). Moreover, no mortality was observed in other experimental groups as well. The administration of free TQ or Lip-TQ at a dose of 40 mg/kg did not induce any significant weight loss on day 8 as compared to day 1 (Table 1).

### 3.7. Administration of Free TQ or Lip-TQ Did Not Induce any Remarkable Changes in Hematological Parameters

The administration of free TQ or Lip-TQ did not induce significant changes in hematological parameters except the erythrocyte count in the mice treated with free TQ at a dose of 40 mg/kg (Table 2). The erythrocyte counts were found to be (8.44 ± 1.1) × 10^6^ in the normal control mice as compared to (6.1 ± 0.76) × 10^6^ per mm^3^ in the blood of the mice injected with free TQ at a dose of 40 mg/kg (*p* < 0.05).

### 3.8. Effect of Free TQ or Lip-TQ on Liver and Kidney Function Parameters

The effect of free TQ or Lip-TQ on liver toxicity was determined by analyzing the levels of AST and ALT, whereas the kidney function was evaluated by the changes in the levels of blood urea nitrogen (BUN) and creatinine (Table 3). The administration of free TQ at a dose of 40 mg/kg increased the level of AST to 69 ± 7.6 IU/L, which was significantly higher as compared to 24 ± 4.6 IU/L in the vehicle control (*p* < 0.01). The mice treated with Lip-TQ (40 mg/kg) exhibited a significant elevation in the AST level to 47.2 ± 7.8 (*p <* 0.05). Similarly to the AST level, the level of ALT was also significantly increased in the mice treated with free TQ (*p <* 0.05).

The level of BUN significantly increased from 18 ± 2.8 to 34.5 ± 6.6 mg/dL in the mice treated with free TQ at a dose of 40 mg/kg (*p <* 0.05), whereas Lip-TQ at a dose of 40 mg/kg did not induce any significant elevation in the BUN level (Table 3). Creatinine, the second important parameter of kidney function, did not induce any significant increase in the mice treated with free TQ or Lip-TQ at a dose of 40 mg/kg (Table 3).

### 3.9. Susceptibility of Mice to Intravenous Infection by A. baumannii

The susceptibility of mice to *A. baumannii* was examined in order to choose an appropriate bacterial inoculum for infection. The mice infected with the largest bacterial inoculum (1 × 10^8^ CFUs) died within four days after the *A. baumannii* infection, whereas all the mice infected with 5 × 10^7^ CFUs died by day 10 post-infection (data not shown). The mice infected with 1 × 10^7^, 5 × 10^6^, 1 × 10^6^ CFUs exhibited a 30%, 80% and 90% survival rate on day 10 post-infection, respectively. Based on the survival results, we selected 1 × 10^7^ CFUs of *A. baumannii* to infect mice in further experiments.

### 3.10. Treatment with Lipsomal-TQ-Protected Mice against the Systemic Infection of Drug-Sensitive A. baumannii

Mice in the saline-treated group or the sham liposome-treated group died within 14 days post-infection with drug-sensitive *A. baumannii* (Figure 8A). The mean survival time (MST) in the saline-treated group was 7.5 days, whereas the mice treated with free TQ (1 mg/kg) had the MST of 11.5 days (*p* = 0.0271). The *A. baumannii*-infected mice treated with Lip-TQ (1 mg/kg) had the MST of 15 days as compared to that of eight days in the mice treated with sham liposomes (*p* = 0.009). The *A. baumannii*-infected mice treated with Lip-TQ at a dose of 5 mg/kg exhibited a 60% survival rate, whereas the mice treated with free TQ at the same dose had a 20% survival rate on day 30 post-treatment (Figure 8A). The mice in the group treated with Lip-TQ (10 mg/kg) had the highest survival rate of 90% in comparison to a 40% survival rate of the group of mice treated with free TQ at the same dose (*p* = 0.0195). The efficacy of treatment was determined by evaluating the bacterial burden in lung tissues (Figure 8B). The mice in the saline-treated group had the bacterial load of 884,628 ± 189,985 CFUs/gm in the lung tissue, whereas free TQ (5 mg/kg and 10 mg/kg) reduced the bacterial load to 296,945 ± 50,277 and 148,027 ± 39,521 CFU/g, respectively (*p <* 0.001). Treatment with Lip-TQ (5 mg/kg and 10 mg/kg) further decreased the bacterial load to 57,749 ± 15,126 and 10,475 ± 6060 CFU/g that was significantly lower as compared to the bacterial load of 905,327 ± 180,579 in the sham-Lip-treated mice (*p <* 0.001). On day 30 post-infection, the surviving mice were sacrificed to assess the residual bacterial infection in lung tissues. The results demonstrated that all the surviving mice were found to be free of the *A. baumannii* infection.

AMX demonstrated a potent antibacterial activity against drug-sensitive *A. baumannii* in a mouse model (Figure 8C). The *A. baumannii*-infected mice treated with AMX (10 mg/kg) exhibited a 100% survival rate on day 30 post-infection (Figure 8C), which is highly significant as compared to the saline-treated mice (*p <* 0.001). The mice treated with AMX at a dose of 5 mg/kg had a 70% survival rate (*p* = 0.0023). The severity of the infection was analyzed by assessing the bacterial load in lung tissues (Figure 8D). The mice in the saline-treated group had a bacterial load of 751,295 ± 150,582 CFU/g in lung tissues (Figure 8D), whereas, the treatment with AMX (5 mg/kg and 10 mg/kg) reduced the bacterial load to 41,749 ± 7987 and 3708 ± 2746 CFU/g, respectively (*p <* 0.001). AMX treatment at a dose of 1 mg/kg reduced the bacterial load to 372,270 ± 78,934 CFU/g of the lung tissue, which was significantly lower as compared to the bacterial load in lung tissues of the saline-treated mice (*p <* 0.01). On day 30 post-infection, the surviving mice were sacrificed to monitor the residual bacterial infection in lung tissues. The results demonstrated that all the mice were found to be free of the *A. baumannii* infection.

### 3.11. Lip-TQ, but Not AMX, Was Effective in the Treatment of the Systemic Infection of Drug-Resistant A. baumannii

The mice in the saline-treated or sham liposome-treated groups died by day 12 post-infection with drug-resistant *A. baumannii* (Figure 9A). Treatment with Lip-TQ (10 mg/kg) imparted the highest survival rate of 70%, whereas the mice treated with free TQ at the same dose had only a 20% survival rate (*p* = 0.0298). In a similar way, the mice in the Lip-TQ (5 mg/kg)-treated group had a 30% survival rate. Contrary to this, the mice treated with free TQ at the same dose died within 30 days post-infection (*p* = 0.0277). The severity of the infection was also analyzed by determining the bacterial load in lung tissues. The number of CFUs of *A. baumannii* was found to be 934,628 ± 208,329 per gram of the lung tissue in the saline-treated group (Figure 9B). The mice treated with Lip-TQ (10 mg/kg) had the lowest bacterial load of 47,061 ± 12,412 CFUs in lung tissues in comparison to 189,027 ± 45,542 CFUs in the lungs of the mice treated with the same dose of free TQ (Figure 9B). The bacterial load was found to be 108,883 ± 33,421 CFU/g of the lung tissue of the mice treated with Lip-TQ (5 mg/kg) as compared to 390,279 ± 16,765 CFUs in the mice treated with free TQ at the same dose. Lip-TQ was most effective at a dose of 10 mg/kg that reduced the bacterial load to 20,394 ± 8,105 CFU/g of the lung tissue (*p <* 0.001).

*A. baumannii* (ATCC 19606) did not respond to AMX treatment in a mouse model. The mean survival time (MST) of the mice in the group treated with AMX at a dose of 10 mg/kg was 8.5 days as compared to that of six days in the saline-treated mice (Figure 9C) (*p >* 0.05). Similarly, the bacterial load was found to be very high in lung tissues of the mice treated with AMX at the doses of 1, 5 and 10 mg/kg (Figure 9D). The bacterial load was found to be 922,051 ± 57,960 per gram of the lung tissue in the saline-treated group (Figure 9D), whereas the bacterial load in lung tissues of the mice treated with AMX at the highest dose of 10 mg/kg was found to be 894,903 ± 168,494 (*p >* 0.05). This shows that AMX treatment did not reduce the severity of the *A. baumannii* infection in the treated mice.

### 3.12. Treatment with Lip-TQ, Not AMX, Alleviated the Parameters of Total Leukocytes and Neutrophils in the Mice Infected with Drug-Resistant A. baumannii

The counts of total leukocytes and neutrophils were significantly increased in the *A. baumannii*-infected mice. The leukocyte count increased from 6277 ± 959 to 17,766 ± 1533 per mm^3^ in the blood of *A. baumannii* infected mice (Figure 10A) (*p <* 0.001). Since TQ at a dose of 10 mg/kg alleviated *A. baumannii* infection, the mice treated with free TQ or Lip-TQ at a dose of 10 mg/kg had significantly reduced leukocyte numbers on day 7 post-treatment (*p <* 0.001). *A. baumannii* infected mice treated with Lip-TQ (10 mg/kg) had 7665 ± 1504 per mm^3^ as compared to the leukocyte count of 17,766 ± 1533 per mm^3^ in the blood of untreated mice (*p <* 0.001). Treatment with free TQ at a dose of 10 mg/kg significantly reduced the leukocyte count to 11,664 per mm^3^ (*p <* 0.01). *A. baumannii*-infected mice treated with AMX exhibited significantly higher levels of leukocytes as the AMX treatment at the given doses was ineffective to eliminate the *A. baumannii* infection (Figure 10B). The leukocyte count increased from 6014 ± 920 to 18,100 ± 1400 per mm^3^ in the blood of the *A. baumannii*-infected mice (Figure 10B). Treatment with AMX at a dose of 10 mg/kg reduced the leukocyte count to 15,652 ± 1459 per mm^3^, which was insignificantly reduced as compared to the leukocyte count in the saline-treated mice (*p >* 0.05).

Neutrophils are important cells of the innate immune response and their numbers increase manifold in acute bacterial infections. The findings of the present study showed that the neutrophil count increased from 645 ± 130 to 2001 ± 330 per mm^3^ of blood in the *A. baumannii*-infected mice (Figure 10C) (*p <* 0.001). Administration of free TQ or Lip-TQ (10 mg/kg) decreased the neutrophil count in the infected mice (*p <* 0.01 and *p <* 0.001, respectively). Noteworthy, the *A. baumannii*-infected mice treated with Lip-TQ (5 mg/kg) had a significantly lower neutrophil count compared to the sham-Lip-treated mice (*p <* 0.05). Similarly to the leukocyte count, the AMX-treated mice had a remarkably increased neutrophil count (Figure 10D). The neutrophil count increased from 636 ± 60 to 2107 ± 312 per mm^3^ of blood in the *A. baumannii*-infected mice (*p <* 0.001). Treatment with AMX at a dose of 10 mg/kg reduced the neutrophil count to 1780 ± 117 per mm^3^ which was insignificantly lower as compared to the neutrophil count in the saline-treated mice (*p >* 0.05).

### 3.13. Treatment with Lip-TQ, Not AMX, Alleviated the Parameters of Inflammation Markers in the Mice Infected with Drug-Resistant A. baumannii

The C-reactive protein (CRP) and procalcitonin (PCT) are common blood biomarkers in bacterial infections. The results of the present study demonstrated that the CRP level was considerably increased in the blood of the *A. baumannii*-infected mice (Figure 11A). The level of the CRP increased to 31.33 ± 4 µg/mL, which was significantly higher as compared to the CRP value of 3 ± 1.8 µg/mL in the uninfected mice (*p <* 0.001). Treatment at a dose of 10 mg/kg of free TQ significantly decreased the CRP level to 10.33 ± 3.5 (*p <* 0.001), whereas treatment with Lip-TQ at the same dose reduced the CRP level to 6.667 ± 2 (*p <* 0.001). On the other hand, the drug-resistant *A. baumannii*-infected mice treated with AMX did not exhibit any significant change in the CRP level (Figure 11B). The mice in the group treated with the highest dose of AMX (10 mg/kg) had 27.67 ± 4.7 µg/mL of the CRP as compared to that of 30.33 ± 5.5 µg/mL in the saline-treated mice (*p >* 0.05).

Similarly to the CRP level, the level of PCT was elevated to 89.67 ± 14.5 pg/mL in the *A. baumannii*-infected mice (Figure 11C), which was significantly higher as compared to 3.6 ± 1.5 pg/mL in the normal control mice (*p <* 0.001). Free TQ at a dose of 10 mg/kg reduced the PCT level to 51 ± 14.93 pg/mL (*p <* 0.05). Importantly, the treatment with Lip-TQ (5 and 10 mg/kg) substantially reduced PCT levels to 39 ± 11.79 pg/mL and 30.67 ± 7 pg/mL, respectively (*p <* 0.01 and *p <* 0.001, respectively). Similarly to its effect on the CRP level, AMX treatment could not alleviate the level of PCT in the *A. baumannii*-infected mice (Figure 11D). The level of PCT was elevated to 104 ± 17.78 pg/mL in the saline-treated mice as compared to 3.67 ± 1.2 pg/mL in the blood of the normal control mice (*p <* 0.001). Treatment of mice with the highest dose of AMX (10 mg/kg) reduced the PCT level from 104 ± 17.78 pg/mL to 94 ± 14.4 pg/mL, which was statistically insignificant (*p >* 0.05).

## 4. Discussion

In recent years, *A. baumannii* has been posing a serious threat to the human population and there is a dire need to find new antibiotics for the treatment [29]. Recently, there has been a huge increase in the emergence of drug-resistant isolates of *A. baumannii*. Multiple antibiotics, including aminopenicillins and first- and second-generation cephalosporins have been shown to be ineffective in the treatment of the drug-resistant *A. baumannii* infection. They have adopted various drug resistance mechanisms, including increased expression of multidrug efflux pumps, target gene mutation, enzymatic inactivation of antibiotics and changes in outer membrane permeability [30]. Keeping in mind an increased emergence of drug-resistant *A. baumannii*, it is important to find an effective and safe antibacterial agent to combat them. The findings of the present study demonstrated that treatment with Lip-TQ was effective against both drug-sensitive and drug-resistant *A. baumannii*.

In the present work, we included both drug-sensitive and drug-resistant *A. baumannii*. *A. baumannii* (ATCC 19606) has exhibited resistance to multiple antibiotics, including ampicillin, amoxicillin, chloramphenicol, cefotaxime, cotrimoxazole, penicillin, erythromycin, clindamycin, cephalothin, ceftazidime and aztreonam. Because of the broad-spectrum antimicrobial activity of TQ, we attempted to analyze the activity of TQ against *A. baumannii*. Interestingly, TQ exhibited antibacterial activity against both drug-sensitive and drug-resistant *A. baumannii*. The in vitro activity of TQ against *A. baumannii* was demonstrated by the agar well diffusion, dilution and time-kill assays. However, the MIC of TQ (5 µg/mL) was found to be higher against drug-resistant *A. baumannii* as compared to MIC of 2 µg/mL against drug-sensitive *A. baumannii*. The results of the time-kill assay demonstrated that TQ killed not only drug-sensitive *A. baumannii*, but also killed more than 99% of drug-resistant *A. baumannii* 24 h after the treatment. However, the bactericidal effect of TQ against drug-resistant *A. baumannii* was observed at a higher concentration. Interestingly, *A. baumannii* did not exhibit any bacterial regrowth in the presence of TQ. The antibacterial activity of TQ is suggested to be mediated through the generation of reactive oxygen species (ROS) [31]. Besides, TQ has been shown to inhibit biofilm formation in some pathogenic bacteria, including *Staphylococcus aureus, Staphylococcus epidermidis, Enterococcus faecalis* and Pseudomonas aeruginosa [32]. A study by Dera et al. demonstrated TQ-induced reduction in the extracellular cell wall of bacteria as revealed by electron microscopy [33]. On the other hand, amoxicillin (AMX) was highly effective against drug-sensitive *A. baumannii*, but not against drug-resistant *A. baumannii* as shown by the agar well diffusion, dilution and time-kill studies.

TQ has been shown to possess therapeutic effects in the treatment of various disease models [7]. However, the clinical use of TQ has been hampered because of its low solubility and bioavailability [19]. In order to increase its therapeutic efficacy, the liposome- and nanoparticle-based formulations of TQ have been shown to possess greater efficacy against various diseases [18,24,34,35,36]. In this study, we prepared TQ-incorporated liposomes to increase the activity and decrease the toxicity of the drug. Phospholipids and cholesterol are safe and are commonly used in the preparation of liposomal antibiotics and vaccines [37]. Liposomes were prepared with DPPC and cholesterol to prepare TQ-loaded liposomes. Being hydrophobic in nature, about 90% of TQ was incorporated in the lipid bilayer of the DPPC liposomes. The stability of the liposomal formulation of TQ was determined in the presence of deionized water or human serum. The findings suggested that the liposomal formulation of TQ was very stable due to a strong interaction between the lipids and TQ due to their hydrophobic nature. In order to evaluate the toxicity of TQ, free or liposomal TQ was administered to mice at the doses of 1, 10, 20 and 40 mg/kg. The mice that were treated with Lip-TQ exhibited comparatively lower toxicity as compared to free TQ at the same dose (Table 1, Table 2 and Table 3). The effect of free TQ or Lip-TQ was analyzed by determining the erythrocyte, leukocyte and platelet counts in the blood of the treated mice. The mice in the group treated with free TQ at a dose of 40 mg/kg exhibited a significantly lower erythrocyte count (Table 2). Biochemical parameters, including AST, ALT, BUN and creatinine, were analyzed in order to assess the toxicity of TQ in terms of the liver and kidney functions. Furthermore, free TQ at a dose of 40 mg/kg caused significant liver and kidney toxicity as compared to the treatment with Lip-TQ at a comparable dose (Table 3). However, the administration of Lip-TQ (40 mg/kg) induced a moderate elevation in the AST level. These findings revealed that the incorporation of TQ in liposomes considerably reduced toxicity of the drug because liposomes slowly release encapsulated drugs into the systemic circulation as compared to the burst release of the free drug. These results are in agreement with the results of earlier studies that demonstrated increased efficacy and reduced toxicity of TQ upon incorporation in liposomes [38].

Because of the recent surfacing of multidrug-resistant isolates of *A. baumannii* and toxicity of antibiotics, there is a need in finding new antibiotics with greater efficacy and decreased toxicity. In the current study, we showed the activity of TQ against *A. baumannii* in both in vitro and in vivo studies. Importantly, TQ exhibited antibacterial activity against multidrug-resistant *A. baumannii* as shown by the agar well diffusion, dilution and time-kill assays and in a murine model of systemic infection. For in vivo studies, the drugs were administered into the mice intraperitoneally. The intraperitoneal route was chosen because it is convenient, easy and less stressful for animals. Furthermore, higher volumes of formulations can be repeatedly administered. Nanoparticle formulations of the drugs have exhibited much higher bioavailability as compared to free drugs when administered intraperitoneally due to decreased clearance [39,40]. The results of in vivo studies were found to be very encouraging as the treatment with Lip-TQ was effective in curing the mice infected with the drug-sensitive and drug-resistant *A. baumannii*. The mice infected with drug-sensitive *A. baumannii* exhibited a 90% survival rate on day 30 after treatment with Lip-TQ (10 mg/kg), whereas the drug-resistant *A. baumannii*-infected mice had a 70% survival rate. Since TQ is poorly soluble in aqueous media, it shows poor efficacy and bioavailability in animal models. It is also substantiated by the results of the present study that demonstrated lower effectiveness of free TQ as compared to Lip-TQ in the treatment of the *A. baumannii* infection. The mice infected with drug-sensitive *A. baumannii* exhibited a 40% survival rate, whereas the mice infected with drug-resistant *A. baumannii* exhibited a 20% survival rate after the treatment with free TQ at a dose of 10 mg/kg. The survival data were also supported by the bacterial load data, which showed the smallest number of CFUs in the mice treated with Lip-TQ at a dose of 10 mg/kg. This suggested that the incorporation of TQ in liposomes increased its efficacy and bioavailability. Lung tissues contain an abundance of reticuloendothelial system (RES) cells. One of the important properties of liposomes is that they are avidly taken by cells of the RES. Thus, liposomes seem to be an important drug delivery system that can be successfully exploited to treat lung infections [41]. AMX exhibits broad-spectrum activity and is one of the most commonly used antibiotics to treat both Gram-positive and Gram-negative bacterial infections. This is also supported by the results of the present study that shows the most reduced bacterial load in lung tissues of the mice treated with Lip-TQ. In the current study, AMX was used as a control antibiotic to treat both drug-sensitive and drug-resistant *A. baumannii* infection in mice. AMX was found to be very effective against drug-sensitive *A. baumannii* and the infected mice treated with AMX at a dose of 10 mg/kg exhibited a 100% survival rate. On the contrary, AMX was ineffective against drug-resistant *A. baumannii* at the same dose. This suggested that TQ, particularly Lip-TQ, may prove an important formulation to treat drug-resistant *A. baumannii*.

The effect of treatment on the alleviation of the *A. baumannii* infection was evaluated by analyzing the total leukocyte and neutrophils counts in the blood, the levels of inflammatory markers such as the CRP and PCT. Leukocytes, particularly neutrophils, are the first responders to acute bacterial infections [42]. Leukocytes and neutrophils feature a sharp increase in acute bacterial infections. The findings of the present study demonstrated that the untreated *A. baumannii*-infected mice exhibited significantly higher leukocyte and neutrophil counts as compared to the uninfected normal control mice. Treatment with Lip-TQ considerably alleviated the *A. baumannii* infection and the treated mice exhibited lower leukocyte and neutrophil counts. Interestingly, neutral liposomes have not been shown to interact with leukocytes and alter their numbers in the systemic circulation [43]. AMX treatment significantly reduced the leukocyte and neutrophil counts in the mice infected with drug-sensitive *A. baumannii*. On the other hand, AMX could not alleviate the leukocyte and neutrophil counts in the mice infected with drug-resistant *A. baumannii*.

The CRP and PCT, blood biomarkers in the diagnosis of bacterial infections, are highly elevated in bacterial infections and inflammations [44]. The findings of the current study showed highly elevated levels of the CRP and PCT in the blood of the *A. baumannii*-infected mice. The *A. baumannii* infection has been shown to induce systemic inflammation by virtue of releasing its outer membrane vesicles [45]. Moreover, the production of proinflammatory cytokines in response to *A. baumannii* further aggravates inflammation in the infected mice [46]. Because of substantial elimination of the *A. baumannii* infection in the Lip-TQ-treated mice, there were reduced levels of the CRP and PCT in the blood of the treated mice. TQ is a well-known anti-inflammatory agent and has a therapeutic role in the treatment of many inflammatory diseases [6,9]. In addition to its antibacterial activity against *A. baumannii*, the anti-inflammatory effect of TQ may have an important role in the mitigation of the CRP and PCT levels. At the same time, the treatment with AMX was ineffective against drug-resistant *A. baumannii* and could not alleviate the blood levels of the CRP and PCT.

## 5. Conclusions

A liposomal formulation of TQ was prepared and evaluated against *A. baumannii* both in vitro and in a mouse model. Lip-TQ exhibited greater activity against the drug-sensitive and drug-resistant *A. baumannii* as compared to the activity of free TQ at the respective doses. The greater activity of Lip-TQ was substantiated by the increased survival rate and reduced bacterial load in lung tissues of the treated mice. Moreover, the leukocyte, neutrophil, CRP and PCT levels were significantly reduced in the *A. baumannii*-infected mice treated with Lip-TQ, particularly at a dose of 10 mg/kg. The greater activity of Lip-TQ is attributed to the property of liposomes to release the incorporated drugs into the blood at the constant and sustained level. Moreover, it also results in the reduced toxicity of the drug. AMX was used as a control drug and was highly effective in the treatment of the drug-sensitive *A. baumannii* infection in mice. However, the treatment with AMX could not cure the drug-resistant *A. baumannii* infection. Collectively, these findings indicated that Lip-TQ may prove to be an effective therapeutic formulation in the treatment of the drug-sensitive or drug-resistant *A. baumannii* infection.

## Figures and Tables

**Figure 1 pharmaceutics-13-00677-f001:**
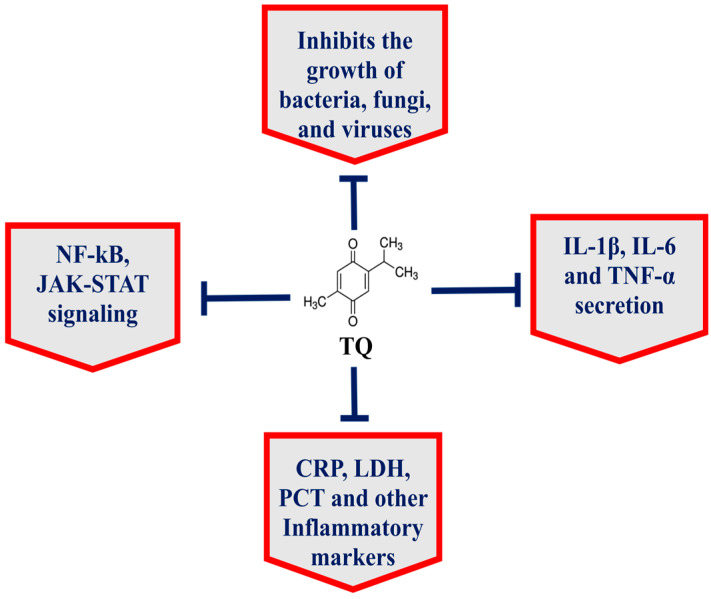
TQ exhibits multitargeting behavior in the treatment of various diseases.

**Figure 2 pharmaceutics-13-00677-f002:**
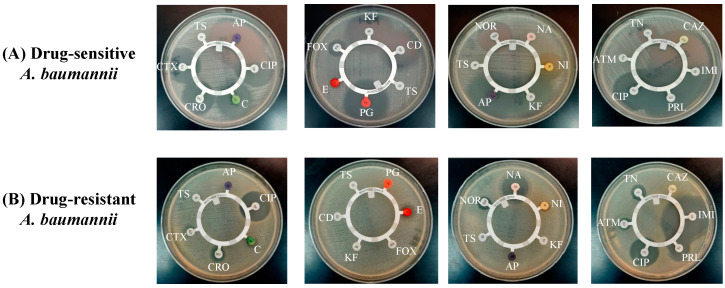
Antibiotic susceptibility screening of (**A**) drug-sensitive and (**B**) drug-resistant *A. baumannii*.

**Figure 3 pharmaceutics-13-00677-f003:**
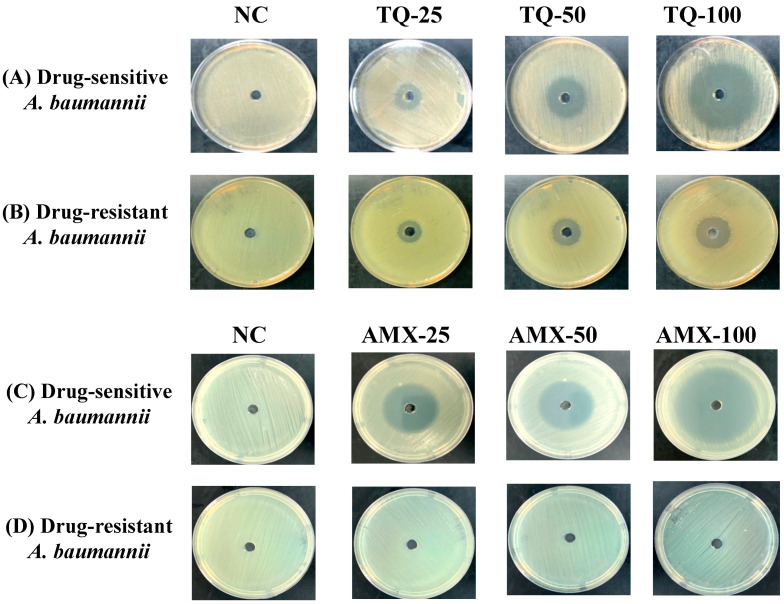
The activity of TQ or AMX against the drug-sensitive and drug-resistant *A. baumannii*. The activity of TQ (25, 50 and 100 µg/mL) against the (**A**) drug-sensitive and (**B**) drug-resistant *A. baumannii*. The activity of AMX (25, 50 and 100 µg/mL) against the (**C**) drug-sensitive and (**D**) drug-resistant *A. baumannii*. Negative controls (NC) contain 5% DMSO.

**Figure 4 pharmaceutics-13-00677-f004:**
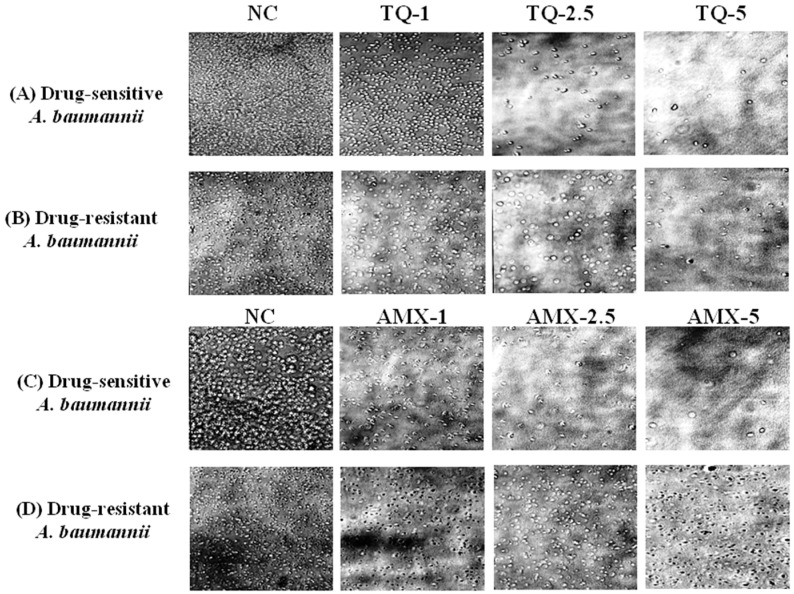
Microscopic analysis of the effect of TQ or AMX against the drug-sensitive and drug-resistant *A. baumannii*. The activity of TQ (1, 2.5 and 5 µg/mL) against the (**A**) drug-sensitive and (**B**) drug-resistant *A. baumannii*. The activity of AMX (1, 2.5 and 5 µg/mL) against the (**C**) drug-sensitive and (**D**) drug-resistant *A. baumannii*. Negative controls (NC) contain 5% DMSO.

**Figure 5 pharmaceutics-13-00677-f005:**
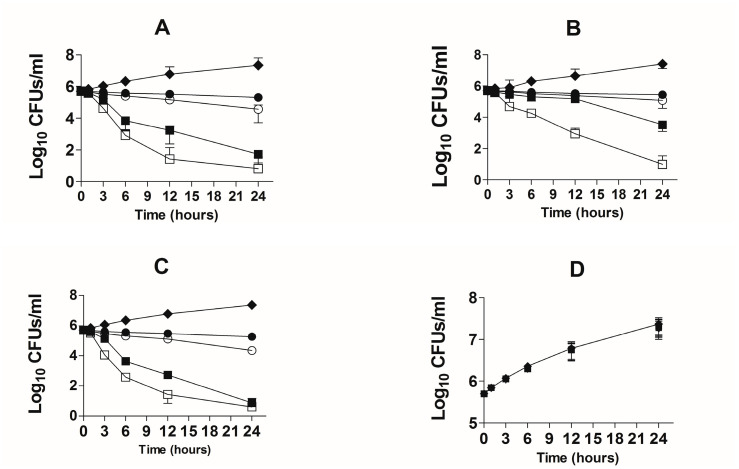
Time-kill curves of TQ or AMX at the doses of 1, 2, 4 and 8 µg/mL against the drug-sensitive and drug-resistant *A. baumanni*. Time-kill curves of TQ against the (**A**) drug-sensitive and (**B**) drug-resistant *A. baumannii*. Time-kill curves of AMX against the (**C**) drug-sensitive and (**D**) drug-resistant *A. baumannii*. The data are represented as the means ± SD of three independent experiments. Control (♦), TQ—1 µg/mL or AMX—1 µg/mL (●), TQ—2 µg/mL or AMX—2 µg/mL (○), TQ—4 µg/mL or AMX—4 µg/mL (■), TQ—8 µg/mL or AMX—8 µg/mL (☐).

**Figure 6 pharmaceutics-13-00677-f006:**
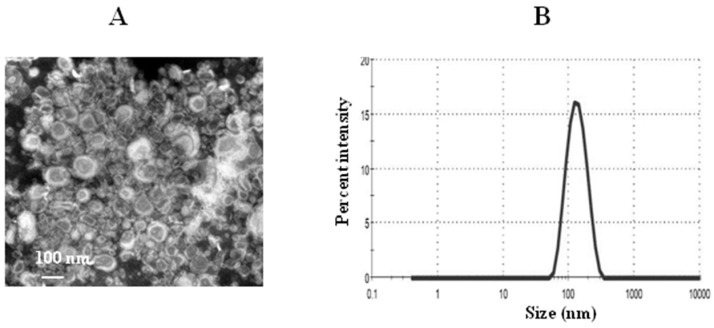
Characterization of liposomes. (**A**) A transmission electron microscopy (TEM) image of TQ liposomes, (**B**) size of TQ liposomes.

**Figure 7 pharmaceutics-13-00677-f007:**
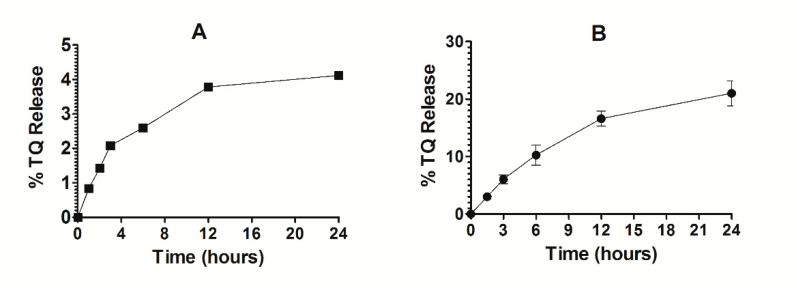
Stability and release kinetics of Lip-TQ. (**A**) Stability of Lip-TQ in deionized water, (**B**) Release kinetics of TQ from the liposomes into the serum.

**Figure 8 pharmaceutics-13-00677-f008:**
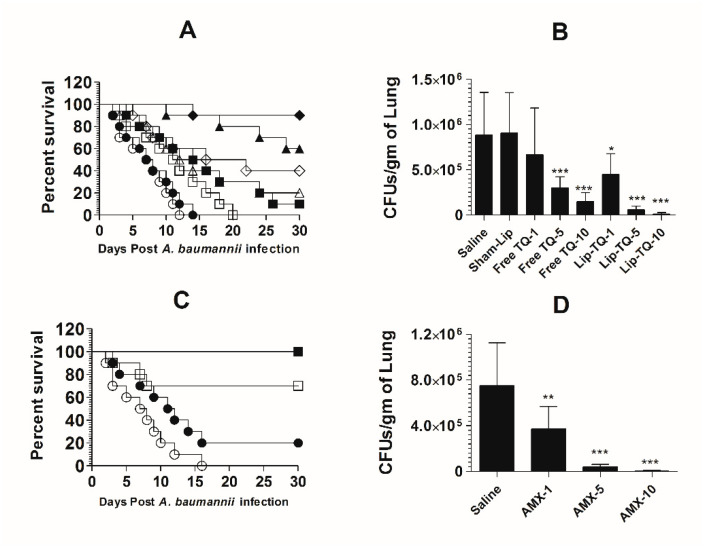
Lip-TQ or AMX effectively eliminated the drug-sensitive *A. baumannii* infection in a murine model. (**A**) Each mouse was infected with 1 × 10^7^ CFUs of *A. baumannii* through an intravenous dose. After 12 h of the infection, the mice were treated with 1, 5 and 10 mg/kg of free TQ and Lip-TQ for seven consecutive days. The mice were monitored for 30 days for the survival. Saline (○), sham-Lip (●), free TQ—1 mg/kg (☐), free TQ—5 mg/kg (△), free TQ—10 mg/kg (◊), Lip-TQ—1 mg/kg (■), Lip-TQ—5 mg/kg (▲), Lip-TQ—10 mg/kg (♦). Saline vs. free TQ—1 mg/kg (*p* = 0.0271), sham-Lip vs. Lip-TQ—1 mg/kg (*p* = 0.009), free TQ—5 mg/kg vs. Lip-TQ—5 mg/kg (*p* = 0.0195), free TQ—10 mg/kg vs. Lip-TQ—10 mg/kg (*p* = 0.0195). (**B**) The bacterial load (CFUs) was determined as described in the methodology section. The data are represented as the means ± SD of three independent experiments. * *p* < 0.05, *** *p* < 0.001. (**C**) The mice infected with drug-sensitive A. baumannii were treated with AMX (1, 5 and 10 mg/kg) for seven days and their survival rate was observed on day 30 after the infection. Saline (○), AMX—1 mg/kg (●), AMX—5 mg/kg (☐), AMX—10 mg/kg (■). Saline vs. AMX—5 mg/kg (*p* = 0.0023), saline vs. AMX—10 mg/kg (*p <* 0.001). (**D**) The bacterial load (CFUs) was determined as described in the methodology section. The data are represented as the means ± SD of three independent experiments. ** *p* < 0.01, *** *p* < 0.001.

**Figure 9 pharmaceutics-13-00677-f009:**
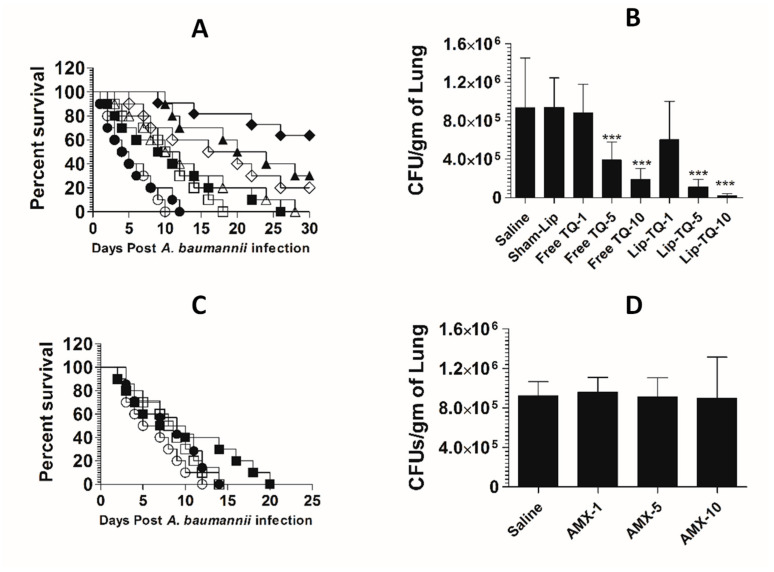
Treatment with Lip-TQ, not with AMX, was effective against the drug-resistant *A. baumannii* infection in a murine model. (**A**) Each mouse was infected with 1 × 10^7^ CFUs of drug-resistant *A. baumannii* and treated with 1, 5 and 10 mg/kg of free TQ or Lip-TQ for seven days as described in the methodology section. The survival rate of the mice was monitored for 30 days. Saline (○), sham-Lip (●), free TQ—1 mg/kg (☐), free TQ—5 mg/kg (△), free TQ—10 mg/kg (◊), Lip-TQ—1 mg/kg (■), Lip-TQ—5 mg/kg (▲), Lip-TQ—10 mg/kg (♦). Saline vs. free TQ—1 mg/kg (*p* = 0.0051), sham-Lip vs. Lip-TQ—1 mg/kg (*p* = 0.0368), free TQ—5 mg/kg vs. Lip-TQ—5 mg/kg (*p* = 0.0272), free TQ—10 mg/kg vs. Lip-TQ—10 mg/kg (*p* = 0.0295). (**B**) Three mice from each group were sacrificed on day 5 after thetreatment and equally weighed portions of the lung tissue were homogenized to determine the bacterial load. The data are represented as the means ± SD of three independent experiments. *** *p* < 0.001. (**C**) The mice infected with drug-resistant A. baumannii were treated with AMX (1, 5 and 10 mg/kg) for seven days and their survival rate was observed on day 30 post-infection. Saline (○), AMX—1 mg/kg (●), AMX—5 mg/kg (☐), AMX—10 mg/kg (■). (**D**) The bacterial load (CFUs) was determined as described in the methodology section. The data are represented as the means ± SD of three independent experiments.

**Figure 10 pharmaceutics-13-00677-f010:**
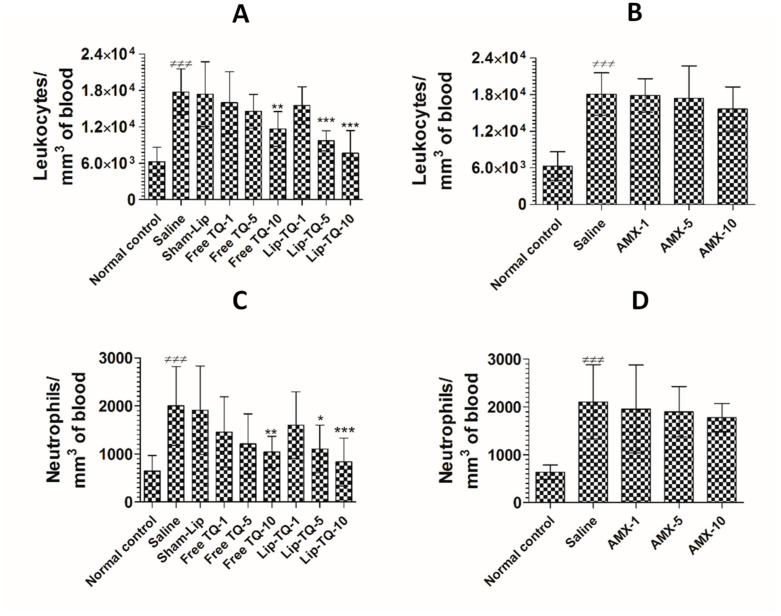
Treatment with Lip-TQ, but not with AMX, alleviated the parameters of total leukocytes and neutrophils in the mice infected with drug-resistant *A. baumannii*. On day 5 post-treatment, the blood was sampled from three mice and the leukocyte and neutrophil counts were analyzed. (**A**) Leukocytes in the free TQ- or Lip-TQ-treated mice, (**B**) leukocytes in the AMX-treated mice, (**C**) neutrophils in the free TQ- or Lip-TQ-treated mice, (**D**) neutrophils in the AMX-treated mice. The data are represented as the means ± SD of three independent experiments. ^≠≠≠^
*p* < 0.001. Normal control vs. saline-treated mice, * *p* < 0.05, ** *p* < 0.001, *** *p* < 0.001. Saline vs. free TQ or Lip-TQ treatment groups.

**Figure 11 pharmaceutics-13-00677-f011:**
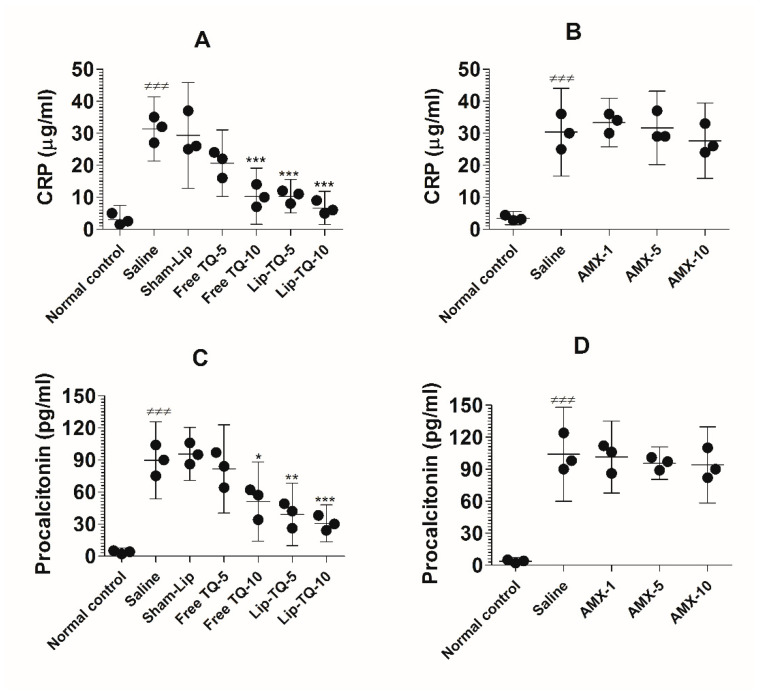
Treatment with Lip-TQ, but not with AMX, alleviated the parameters of the CRP and PCT in the mice infected with drug-resistant *A. baumannii*. On day 5 post-treatment, the blood was sampled from three mice and the levels of the CRP and PCT were analyzed. (**A**) The CRP in the free TQ- or Lip-TQ-treated mice, (**B**) the CRP in the AMX-treated mice, (**C**) PCT in the free TQ- or Lip-TQ-treated mice, (**D**) PCT in the AMX-treated mice. The data are represented as the means ± SD of three independent experiments. ^≠≠≠^
*p* < 0.001. Normal control vs. saline-treated mice, * *p* < 0.05, ** *p* < 0.001, *** *p* < 0.001. Saline vs. free TQ or Lip-TQ treatment groups.

**Table 1 pharmaceutics-13-00677-t001:** Mortality and change in body weight in the mice injected with various doses of free TQ or Lip-TQ. The data are represented as the means ± SD of three independent experiments.

Group	Mortality	Body Weight		
		Day 1	Day 5	Day 11
Vehicle control	0/8	28 ± 2.4	28.8 ± 2.2	29.2 ± 2.1
Sham liposomes	0/8	27.4 ± 3.0	27.7 ± 3.1	28.2 ± 3.3
Free TQ—1 mg/kg	0/8	29.6 ± 1.8	29.6 ± 2.2	30 ± 2.0
Free TQ—10 mg/kg	0/8	28.8 ± 2.5	28.7 ± 2.4	28 ± 3.4
Free TQ—20 mg/kg	0/8	28.6 ± 2.4	28.4 ± 3.6	27.8 ± 3.6
Free TQ—40 mg/kg	0/8	28.2 ± 2.2	26.5 ± 3.4	26.4 ± 3.1
Lip-TQ—1 mg/kg	0/8	26.6 ± 1.7	29.2 ± 1.8	27.7 ± 1.5
Lip-TQ—10 mg/kg	0/8	28.5 ± 3.8	28.6 ± 2.7	27.6 ± 2.5
Lip-TQ—20 mg/kg	0/8	29.4 ± 2.6	29 ± 3.2	28.2 ± 4.0
Lip-TQ—40 mg/kg	0/8	29 ± 3.5	28.4 ± 3.0	27.8 ± 3.8

**Table 2 pharmaceutics-13-00677-t002:** Hematological parameters of the mice injected with various doses of free TQ or Lip-TQ.

Group	Erythrocytes/mm^3^ × (10^6^)	Leukocytes/mm^3^ × (10^3^)	Platelets/mm^3^ × (10^5^)
Vehicle control	8.44 ± 1.1	6.6 ± 2.23	4.64 ± 1.12
Sham liposomes	8.28 ± 1.4	6.8 ± 2.52	4.46 ± 2.04
Free TQ—1 mg/kg	8.12 ± 0.92	7.1 ± 2.54	4.48 ± 1.24
Free TQ—10 mg/kg	7.88 ± 1.6	6.8 ± 1.88	4.59 ± 2.10
Free TQ—20 mg/kg	7.22 ± 0.96	6.6 ± 1.92	4.2 ± 1.66
Free TQ—40 mg/kg	6.1 ± 0.76 *	5.7 ± 1.75	3.8 ± 1.62
Lip-TQ—1 mg/kg	8.76 ± 1.5	7.22 ± 1.72	4.88 ± 1.78
Lip-TQ—10 mg/kg	8.26 ± 1.8	6.94 ± 1.56	4.06 ± 1.63
Lip-TQ—20 mg/kg	7.52 ± 0.92	6.2 ± 2.8	4.1 ± 1.92
Lip-TQ—40 mg/kg	7.33 ± 1.2	6.24 ± 2.4	4.1 ± 1.66

* *p* < 0.05 as compared to vehicle control.

**Table 3 pharmaceutics-13-00677-t003:** Effect of free TQ or Lip-TQ on hepatic and renal toxicity.

Group	AST (IU/L)	ALT (IU/L)	BUN (mg/dL)	Creatinine (mg/dL)
Vehicle control	24 ± 4.6	32 ± 5.8	18 ± 2.8	0.46 ± 0.08
Sham liposomes	26 ± 3.2	27.8 ± 6.2	22 ± 2.2	0.52 ± 0.10
Free TQ—1 mg/kg	32 ± 2.5	28.4 ± 5.2	25 ± 2.8	0.48 ± 0.12
Free TQ—10 mg/kg	36 ± 6.6	27.8 ± 6.2	28 ± 5.1	0.55 ± 0.11
Free TQ—20 mg/kg	48 ± 11.4	32.6 ± 6.0	32.3 ± 4.8	0.64 ± 0.24
Free TQ—40 mg/kg	69 ± 7.6 **	48.8 ± 5.2 *	34.5 ± 6.6 *	0.66 ± 0.16
Lip-TQ—1 mg/kg	22 ± 2.7	26.2 ± 5.4	21 ± 4.4	0.38 ± 0.07
Lip-TQ—10 mg/kg	27.4 ± 4.2	29.4 ± 4.8	22 ± 3.8	0.47 ± 0.11
Lip-TQ—20 mg/kg	36.6 ± 7.4	34.2 ± 7.6	23.6 ± 4.0	0.49 ± 0.09
Lip-TQ—40 mg/kg	47.2 ± 7.8 *	38.2 ± 7.3	25.8 ± 3.8	0.58 ± 0.14

* *p* < 0.05, ** *p* < 0.01 as compared to vehicle control.

## Data Availability

All relevant data have been provided within the manuscript. There are no supporting files and no data was held.

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
