# Peer review of "Safety and Therapeutic Efficacy of Thymoquinone-Loaded Liposomes against Drug-Sensitive and Drug-Resistant Acinetobacter baumannii"

_pharmaceutics, 2021, doi:10.3390/pharmaceutics13050677_

Round 1

Reviewer 1 Report

In this manuscript was developed a liposomal formulation of Thymoquinone (TQ) and was evaluated against A. baumannii both in vitro and in a mouse model by Masood Alam Khans group. The results of this investigation may have various pharmacological and biomedical applications.

Advantages:

The experimental results are presented in detail through figures and graphs and fully support the discussion. The discussion gave a complete answer to the set goal from the introduction. These researches are complex and include experiments in physical pharmacy and experimental pharmacology.

Lack:

In the introductory part, the authors could give a figure with the chemical formula of thymoquinone as well as the scheme of action of thymoquinone (cell signaling pathways and mechanism of action on microorganisms). The goal in the introduction should be emphasized more. The conclusion should be clearly separated from the discussion - a new chapter. Although the manuscript is richly illustrated, it is necessary to improve the resolution of the images.

Author Response

Responding to reviewer 1

We thank the respected reviewer for providing the comments for this manuscript. His comments will surely help to improve the quality of the manuscript.

Comment: In this manuscript was developed a liposomal formulation of Thymoquinone (TQ) and was evaluated against A. baumannii both in vitro and in a mouse model by Masood Alam Khans group. The results of this investigation may have various pharmacological and biomedical applications.

The experimental results are presented in detail through figures and graphs and fully support the discussion. The discussion gave a complete answer to the set goal from the introduction. These researches are complex and include experiments in physical pharmacy and experimental pharmacology.

Response: We are thankful to the honorable reviewer for his encouraging words about the current study.

Comment: In the introductory part, the authors could give a figure with the chemical formula of thymoquinone as well as the scheme of action of thymoquinone (cell signaling pathways and mechanism of action on microorganisms). The goal in the introduction should be emphasized more. The conclusion should be clearly separated from the discussion - a new chapter. Although the manuscript is richly illustrated, it is necessary to improve the resolution of the images.

Response: As per the suggestion of the respected reviewer, we are providing schematic figure for the mechanism of action thymoquinone. We are also improving the introduction part in the revised manuscript. As suggested by the reviewer, conclusion part has been updated. We understand the concern of the reviewer and are improving the quality of the images in the revised manuscript.

Reviewer 2 Report

In this work, Khan and coworkers report the antibacterial activity of liposomal thymoquinone against the drug-sensitive and drug-resistant A. baumannii. Although encapsulating thymoquinone inside liposome is not novel, I found this work very interesting and significant due to promising in-vivo data. 70% survival rate was found in the group treated with liposomal thymoquinone, while standard antibiotic amoxicillin is ineffective. Bacterial load in animal lung and inflammation markers in the blood were also reduced. The results are very promising and certainly advance the field of antibiotic development. Therefore, the work has merit to be published on Pharmaceutics after revisions as noted below.

  1. Figure 3-9 are very blurry. The figure quality must be improved.
  2. The statement “These findings revealed that the incorporation of TQ in liposomes considerably reduced its toxicity in mice” needs an explanation. Why liposomes considerably reduced its toxicity in mice?
  3. The liposome was prepared using DPPC and cholesterol. The authors should add a discussion about PEGylation the liposome to increase circulation and citing a relevant reference (doi.org/10.3390/pharmaceutics12111068)
  4. The authors should also explain in more details the discussion: “Contrary to Lip-TQ, free TQ was less effective to cure A. baumannii in fection.”
  5. In section 3.11 Treatment with Lip-TQ, not AMX, alleviated the parameters of total leukocytes and neutrophils in mice infected with the drug-resistant A. baumannii, the authors should add a note that it has been found that liposomes do not associate and increase the numbers of leukocytes and neutrophils citing the relevant work (doi.org/10.1002/smll.202002861).
  6. What is the charge (zeta potential) of the liposomal thymoquinone?
  7. The sentence “DPPC phospholipids are considered the safe lipids and are commonly used in the preparation of drug-loaded liposomes” should be revised as “Phospholipids and cholesterol are safe and commonly used in the preparation of liposomal antibiotics and vaccines” because cholesterol was also used in this liposome formulation. The revised sentence should cite a relevant reference: doi.org/10.1002/adhm.202002142
  8. The authors should clarify how free thymoquinone was injected while it has low solubility in water. If the highest dose of 40mg/kg was still below the solubility of thymoquinone, a higher dose should be tried to leverage the enhanced solubility of liposomal thymoquinone.

Author Response

Response to reviewer 2

Comment: Figure 3-9 are very blurry. The figure quality must be improved.

Response: As suggested by the respected reviewer, we are improving the quality of the figures in the revised manuscript.

 Comment: The statement “These findings revealed that the incorporation of TQ in liposomes considerably reduced its toxicity in mice” needs an explanation. Why liposomes considerably reduced its toxicity in mice?

Response: We have provided an explanation to support this statement.

 Comment: The liposome was prepared using DPPC and cholesterol. The authors should add a discussion about PEGylation the liposome to increase circulation and citing a relevant reference (doi.org/10.3390/pharmaceutics12111068)

Response: As suggested by the reviewer, we have added the suggested reference in the introduction part.

Comment: The authors should also explain in more details the discussion: “Contrary to Lip-TQ, free TQ was less effective to cure A. baumannii infection.”

Response: We agreed with the reviewer and included the explanation to support the above mentioned statement.

Comment: In section 3.11 Treatment with Lip-TQ, not AMX, alleviated the parameters of total leukocytes and neutrophils in mice infected with the drug-resistant A. baumannii, the authors should add a note that it has been found that liposomes do not associate and increase the numbers of leukocytes and neutrophils citing the relevant work (doi.org/10.1002/smll.202002861).

Response: We have included the suggested reference in the discussion section of the revised manuscript.

Comment: What is the charge (zeta potential) of the liposomal thymoquinone?

Response: The zeta potential of thymoquinone liposomes was found to be 21 mg. We are mentioning it in the text of the revised manuscript.

Comment: The sentence “DPPC phospholipids are considered the safe lipids and are commonly used in the preparation of drug-loaded liposomes” should be revised as “Phospholipids and cholesterol are safe and commonly used in the preparation of liposomal antibiotics and vaccines” because cholesterol was also used in this liposome formulation. The revised sentence should cite a relevant reference: doi.org/10.1002/adhm.202002142.

Response: We have revised the sentence according to the suggestion of the reviewer. The suggested reference has also been cited in the revised manuscript.

Comment: The authors should clarify how free thymoquinone was injected while it has low solubility in water. If the highest dose of 40 mg/kg was still below the solubility of thymoquinone, a higher dose should be tried to leverage the enhanced solubility of liposomal thymoquinone.

Response: Thymoquinone was dissolved in DMSO and was diluted in normal saline to have 1% DMSO in the final solution.

Reviewer 3 Report

In this study, the authors have evaluated the pharmacological properties of liposomal thymoquinone, which have been developed previously. There are almost no new data in terms of pharmaceutics. The authors should submit the manuscript to journals in the field of pharmacology.

Author Response

Response to reviewer 3

Comment: In this study, the authors have evaluated the pharmacological properties of liposomal thymoquinone, which have been developed previously. There are almost no new data in terms of pharmaceutics. The authors should submit the manuscript to journals in the field of pharmacology.

Response: We respect the opinion of the honorable reviewer. The current manuscript contains the data of drug dosage and designing of drug formulations, which are well within the scope of “Pharmaceutics”.

Round 2

Reviewer 2 Report

The previous comments have been sufficiently addressed. Congratulation to the authors on the good work.

Author Response

We thank the respected reviewer for approving our responses to the comments.

Reviewer 3 Report

  1. In this study, the author test the efficacy and safety of liposomal thymoquinone against A. baumannii. They claimed the data were well within the scope of “Pharmaceutics”. But the formulation is not novel because they have already published the same formulation. Again, what is the novel points as pharmaceutics? I think only release profiles added during revision was meaningful as the study of pharmaceutics. It is better to perform the pharmacokinetics study.
  2. In line 80, the author claimed the formulation is novel. I think the formulation was traditional neutral liposomes (DPPC/Chol).
  3. The liposomal thymoquinone was administered via intraperitoneal route. It is difficult to inject the formulation via intraperitoneal route in human, especially about repeated administration. Administration routes affects the biodistribution largely. How to interpret the murine data into human?
  4. Why was the zeta potential strongly positive?
  5. The solvent for liposomes was PBS. Then, the formulation seems to be unstable for long-term storage. The authors should test the stability of the liposomes.
  6. Please indicate the temperature during preparation of the liposomes (hydration, extrusion). Temperature is very important factor to reproduce the formulation.
  7. I think it is impossible to precipitate the liposomes by usual centrifugation. It should require ultracentrifugation.
  8. It is better to discuss the mechanism of action of thymoquinone against A. baumannii.
  9. The unit ‘rpm’ for centrifugation should be converted into centrifugal force (g).
  10. Please specify the institutional approval number of animal experiments.
  11. Please specify the kind of post-hoc test after ANOVA.
  12. In line 270, Figure 1B may be 2B.
  13. In figure 5, the label (A-B) of the panels were missing.

Author Response

We are highly thankful to the respected reviewer for providing comments for the present manuscript. These comments are very helpful and certainly help to improve the quality of the manuscript.

 Comment: In this study, the author test the efficacy and safety of liposomal thymoquinone against A. baumannii. They claimed the data were well within the scope of “Pharmaceutics”. But the formulation is not novel because they have already published the same formulation. Again, what is the novel points as pharmaceutics? I think only release profiles added during revision was meaningful as the study of pharmaceutics. It is better to perform the pharmacokinetics study.

Response: We agree with the reviewer that performing pharmacokinetic studies will improve the impact of the study. We feel sorry that at this point, we will not able to perform a pharmacokinetics study.  

Comment: In line 80, the author claimed the formulation is novel. I think the formulation was traditional neutral liposomes (DPPC/Chol).

Response: As per the suggestion of the reviewer, we are editing the sentence.

Comment: The liposomal thymoquinone was administered via intraperitoneal route. It is difficult to inject the formulation via intraperitoneal route in human, especially about repeated administration. Administration routes affects the biodistribution largely. How to interpret the murine data into human?

Response: We agree with the respected reviewer that it is very difficult to inject the formulation in human beings through the intraperitoneal route. In order to reduce the sufferings, we selected the intraperitoneal route for the administration of formulation in mice because repeated injection of formulations in mice through the intravenous route is difficult and painful.  

Comment: Why was the zeta potential strongly positive?

Response: We feel very sorry for this error. The zeta potential of the liposomes is 2.1 mV. The DPPC liposome has a zwitterionic nature, and its polar heads may reorient depending on the ionic strength and the presence of encapsulated molecules. Sham DPPC-liposomes present a small negative potential, whereas the addition of thymoquinone changes its zeta potential.

Comment: The solvent for liposomes was PBS. Then, the formulation seems to be unstable for long-term storage. The authors should test the stability of the liposomes.

Response: We reconstituted the liposomes in PBS for the purpose of in vivo use. However, this formulation in water was also equally stable.  

Comment: Please indicate the temperature during preparation of the liposomes (hydration, extrusion). Temperature is very important factor to reproduce the formulation.

Response: The preparation of liposomes, including formation of lipid film, the hydration and extrusion were performed at 370C. As suggested by the respected reviewer, we mentioning this information in the revised manuscript.

Comment: I think it is impossible to precipitate the liposomes by usual centrifugation. It should require ultracentrifugation.

 Response: Liposomes can be settled down by centrifuging at 15,000 rpm for 15 to 30 minutes.

Comment: It is better to discuss the mechanism of action of thymoquinone against A. baumannii.

Response: It is a very good suggestion of the reviewer. We have added some points to discuss the mechanism of action of TQ against bacteria.

Comment: The unit ‘rpm’ for centrifugation should be converted into centrifugal force (g).

 Response: As per the suggestion of the respected reviewer, we are converting the unit of centrifugation from the “rpm” into “g”

Comment: Please specify the institutional approval number of animal experiments.

Response: We have included the institutional approval number in the revised manuscript.

Comment: Please specify the kind of post-hoc test after ANOVA.

Response: ANOVA was followed by a Bonferroni post-test.

Comment: In line 270, Figure 1B may be 2B.

Response: We feel sorry and thank the reviewer for pointing out this mistake.

Comment: In figure 5, the label (A-B) of the panels were missing.

Response: We again thank the respected reviewer for this correction.

Round 3

Reviewer 3 Report

The explanations against my comments are not satisfactory.

Even if the authors cannot perform additional experiments, the authors should explain carefully. Especially, discussion about the administration route is necessary. And, the stability issue is important in the field of pharmaceutics. At least, the authors should check and show the stability data.

Author Response

Response to reviewer 3 comments

Comment: Even if the authors cannot perform additional experiments, the authors should explain carefully. Especially, discussion about the administration route is necessary.

Response: In the discussion section, we have provided the justification for using the intraperitoneal route of the drug administration.

Comment: And, the stability issue is important in the field of pharmaceutics. At least, the authors should check and show the stability data.

Response: We have provided the stability data of TQ liposomes in the Figure 7A of the revised manuscript .